# Efficacy of Expanded Hemodialysis and Comparison with Standard Hemodialysis and Online Hemodiafiltration

**DOI:** 10.3390/ijms26125747

**Published:** 2025-06-16

**Authors:** Adamantia Bratsiakou, Marios Papasotiriou, Panagiota Davoulou, Georgia Andriana Georgopoulou, Evangelos Papachristou, Dimitrios S. Goumenos

**Affiliations:** Department of Nephrology and Kidney Transplantation, University Hospital of Patras, School of Medicine, University of Patras, 26504 Patras, Greece; mpr.adam@hotmail.gr (A.B.); mpapasotir@upatras.gr (M.P.); pdavoulou@gmail.com (P.D.); georgop91@gmail.com (G.A.G.); epapachr@upatras.gr (E.P.)

**Keywords:** expanded hemodialysis, hemodialysis, hemodiafiltration, medium-molecular-weight substances

## Abstract

Patients on standard hemodialysis (HD) show insufficient clearance of medium-molecular-weight uremic toxins, resulting in long-term complications. In this study we investigated the effectiveness of expanded HD (xHD) in the clearance of medium-molecular-weight uremic substances. This prospective study included patients on standard thrice-weekly HD. Participants were treated with xHD for 4 weeks, and the clearance of urea and β2-microglobulin was measured at the beginning and at the end of the study and compared with standard HD (sHD). Moreover, we investigated the clearance of Su-PAR, MCP-1, and activin, comparing sHD, xHD, and online hemodiafiltration (HDF). xHD had the same efficiency in the removal of low-molecular-weight substances compared to sHD but led to a significant decrease in β2-microglobulin levels from the first application of the method (sHD: from 36.9 ± 10.6 to 39.7 ± 18.9 mg/dL, *p* = 0.595 vs. 1st xHD: from 40 (36.5, 41.8) to 11 (9.8, 13.2) mg/dL, *p* = 0.008 vs. after 4 weeks on xHD: from 31.5 (28.5, 34.5) to 8.7 (8.2, 9.0) mg/dL, *p* = 0.008). Moreover, pre-session β2-microglobulin levels significantly decreased after 4 weeks on xHD. Su-PAR, MCP-1, and activin during xHD were also significantly reduced. xHD leads to a significant and cumulative reduction in medium-molecular-weight uremic toxins compared to standard HD.

## 1. Introduction

In recent decades, hemodialysis (HD) has been refined, leading patients with end-stage kidney disease (ESKD) to better quality of life and longer survival compared to the past [1]. However, the morbidity and mortality of patients with ESKD on maintenance dialysis still remain high [2]. Most patients receiving HD are usually hospitalized more than once per year, mainly because of cardiovascular complications and infections, which are the leading causes of death in this population [3,4,5]. The high mortality rate indicates that HD remains a suboptimal treatment, as it fails to remove the full spectrum of uremic toxins. Standard HD clears uremic toxins, primarily through diffusion, and clearance is inversely proportional to the radius of the toxin molecule. As a result, conventional HD fails to completely correct the uremic milieu as it is very effective for low-molecular-weight uremic toxins such as urea, but the clearance of large and protein-bound molecules is suboptimal.

Nowadays, it is known that medium-molecular-weight uremic toxins are accumulated in ESKD and can contribute to the appearance of many toxic effects such as amyloidosis, chronic inflammation, cardiovascular disease, mineral-bone disease, and neuropathy, virtually covering all organ systems [6,7]. These complications, despite increased treatment adequacy, as expressed with optimal Kt/V values, have been correlated with insufficient clearance of uremic toxin with molecular weight between 5 and 50 kDa [8]. In recent years, the development of online hemodiafiltration (HDF) has improved the clearance of such substances and has led to better survival rates and fewer complications [9,10]. In online HDF, large volumes of sterile replacement fluid are obtained by filtration of standard dialysate through a series of filters that retain bacteria and endotoxins, producing ultrapure replacement fluid. Despite the advantages of online HDF, this method has limitations, as it is much more expensive, requires excellent functioning vascular access, and requires dedicated technology [11]. Moreover, the use of high cut-off membranes can lead to excessive loss of useful substances such as albumin [12].

xHD is a new method that does not require special equipment or software but uses a new class of membranes called medium cut-off. Compared with online HDF, xHD does not need replacement fluid. These new membranes are characterized by a smaller inner diameter, which allows high rates of both internal filtration and back-filtration. According to recent studies, this method provides much better clearance than standard HD and similar results with online HDF [13,14]. The aim of the present study was to investigate the efficacy of xHD in the clearance of medium-molecular-weight uremic toxins compared to online HDF and standard HD.

## 2. Results

The baseline demographic and clinical characteristics of the study population are presented in Table 1. In the first part of the study, seven patients (three women) with a mean age of 64.2 ± 10.8 years and a mean previous duration on standard HD of 10.8 ± 4.6 years were included in the study. All the participants were previously treated with standard HD with low-flux membranes and had satisfactory dialysis adequacy with URR (urea reduction ratio) = 72.4 ± 8.5%. The xHD treatment had the same efficiency in the removal of low-molecular-weight substances (urea) as standard HD (Figure 1).

Concerning the removal of medium-molecular-weight substances, standard HD had no effect, whereas HD led to a significant decrease in β2-microglobulin levels during the HD session, even from the first application of the method (standard HD: from 36.9 ± 10.6 to 39.7 ± 18.9 mg/dL, *p* = 0.595 vs. 1st xHD: from 40 (36.5, 41.8) to [11 (9.8, 13.2)] mg/dL, *p* = 0.008 vs. after 4 weeks on xHD: from 31.5 (28.5, 34.5) to 8.7 (8.2, 9.0) mg/dL, *p* = 0.008) (Figure 2). Moreover, after 4 weeks on xHD, the levels of β2-microglobulin before the HD session were significantly lower than the ones at the beginning of the study (Figure 3).

In the second part of the study, we compared the difference in clearance of Su-PAR, MCP-1, and activin-A between 10 patients on standard HD, 10 patients on online HDF, and 10 patients on xHD. Accordingly, xHD was effective in the reduction in these medium-molecular-weight substances. In more detail, Su-PAR was reduced from 7905.6 ± 2083.0 to 7102.4 ± 1907.0 pg/mL (*p* = 0.002), MCP-1 from 602.8 (457.3, 746.3) to 410.3 (311.2, 559.0) pg/mL (*p* = 0.002), and activin-A from 896.2 ± 254.2 to 716.5 ± 158.6 pg/mL (*p* = 0.020) (Figure 4). Patients treated with online HDF showed lower total levels of Su-PAR and MCP-1 after treatment, though this reduction was not significant (Su-PAR: from 6104.1 ± 1783.0 to 5520.5 ± 1121.0 pg/mL, *p* = 0.164; MCP-1: from 518.7 ± 130.4 to 500.1 ± 185.4 pg/mL, *p* = 0.683). As for activin-A, there was an increase at the end of the online HDF session from 707.5 (536.4, 927.3) to 745.3 (491.7, 1594.0) pg/mL, but the difference was again not significant (*p* = 0.432) (Figure 5). Finally, patients treated with standard HD with low-flux membranes had no difference in Su-PAR and MCP-1 during the HD session (Su-PAR: from 7214.4 ± 1403.0 to 7172.2 ± 1531.0 pg/mL, *p* = 0.756; MCP-1: from 578.0 (524.3, 928.1) to 538.8 (442.7, 774.5) pg/mL, *p*= 0.275), whereas we found a significant increase in activin (from 821.3 ± 239.8 to 950.0 ± 294.5 pg/mL, *p* = 0.023) (Figure 6).

## 3. Discussion

The main finding of this study confirmed that xHD leads to a significant and cumulative reduction in the levels of medium-molecular-weight uremic toxins compared to standard HD. This observation is of high clinical importance, as xHD could increase the clearance of medium-molecular-weight toxins, improving the clinical outcomes in patients receiving HD. Moreover, compared to online HDF, xHD seems to be less expensive, as MCO membranes do not require specific HD machines or specific software. This means that these new membranes provide a reliable and cheap HD method, as online HDF cannot be implemented in all ESKD patients.

As already established, the incidence of cardiovascular disease in dialysis patients is increased and is possibly associated with the insufficient clearance of medium-molecular-weight uremic and inflammatory solutes. Efficient renal replacement therapies that could remove these toxins could potentially reduce the high incidence of morbidity and mortality in this group of patients. Peter J. Blankestijn et al. compared the effect of hemodiafiltration or hemodialysis on mortality. In this multinational, randomized, controlled trial, 683 patients were randomized to receive high-dose hemodiafiltration and 677 to receive high-flux hemodialysis. According to the results, the use of high-dose hemodiafiltration resulted in a lower risk of death from any cause than conventional high-flux hemodialysis (hazard ratio, 0.77; 95% confidence interval, 0.65 to 0.93) [15]. Zickler et al. investigated the reduction in inflammatory solutes using MCO membranes in comparison to high-flux HD. According to this study, MCO dialyzers modulate inflammation in chronic HD patients to a greater extent compared to high-flux dialyzers [16]. They measured the levels of TNF-a and IL-6 after four weeks of MCO and found that the levels were significantly lower than in patients on high-flux HD. Furthermore, Zhao et al. published a systematic review and meta-analysis of 18 trials including 853 HD patients. Accordingly, xHD increased the reduction in β2-microglobulin compared to high-flux HD. On the other hand, the reduction ratio of β2-microglobulin in the xHD was lower than the reduction in HDF [14]. These results agree with the results of our study, as we noticed that xHD is better than classic HD in the reduction in medium-molecular-weight substances, but a longer period of xHD application is necessary to compare the results of xHD and online HDF.

Eiamcharoenying et al. compared the effectiveness of xHD and mixed-dilution online HDF in removing middle-molecule uremic toxins. Fourteen HD patients participated in their study and were randomized into two groups. The first group was at first treated with xHD and later with online HDF and vice versa. According to their results, xHD provides similar effectiveness in reducing uremic toxins and could be a good alternative to mixed dilution online HDF [17]. In our study we were able to show that inflammatory molecules such as MCP-1 and markers of increased mortality, such as Su-PAR, in patients with ESKD, were steadily and significantly decreased after consecutive sessions [18,19,20]. This finding could potentially have meaningful clinical implications, as decreased pro-inflammatory molecules and Su-PAR levels could be associated with a reduction in atherosclerotic events and overall mortality in long-term prospective studies.

Moreover, Reque et al. compared xHD and online HDF in the clearance of small- and medium-size molecules. Their study included eight patients who were divided into two groups, where the first one was treated with xHD and then with online HDF while the second group was treated with online HDF at the beginning and xHD after that. According to their results, urea and β2-microglobulin had almost the same reduction rate, but as far as myoglobulin is concerned, the reduction rate with xHD was higher. This means that xHD and online HDF have similar results in the clearance of low- and medium-molecular-weight substances, but xHD is probably better in the clearance of large, medium molecules [21].

Our study has strengths and limitations. The main limitation is that the duration of the intervention was 4 weeks, which can be considered short, as long-term complications like cardiovascular disease and chronic inflammation require prolonged observation to draw safe conclusions; longer interventions are necessary to further delineate the observed effects. Moreover, the sample size was small, and this could have an impact on the robustness of our results. Further randomized trials with larger sample sizes are necessary to clarify if the conclusions of our study could be applied to a wider population. Furthermore, in our study we did not follow a randomized design. A possible alternative could be a randomized cross-over design, but in that case, large wash-out periods would be needed to eliminate possible carry-over effects. In order to overcome this problem in study design, we carefully selected patients with very similar characteristics, like age, dialysis vintage, type of vascular access, and comorbidities (diabetes mellitus, hypertension, heart failure, coronary artery disease), so we believe that the bias is negligible. Despite the limitations, this study allowed us to confirm that xHD is much better than standard HD with low-flux membranes in reducing medium-molecular-weight uremic toxins, and it could reduce the incidence of morbidity and mortality in HD patients by improving the clearance of these uremic toxins. As for online HDF, it seems that xHD could be a reliable alternative method, with adequate results and less cost, as xHD does not require specific equipment and a large volume of ultrapure dialysis replacement fluid; however, this should be confirmed with cost-effectiveness studies [22].

## 4. Materials and Methods

### 4.1. Study Population

This was a prospective, non-randomized, open-label interventional study conducted in the Hemodialysis Unit of the University Hospital of Patras, Greece, between May and December 2022. For the first part of our study, inclusion criteria were (i) age >18 years, (ii) ESKD treated with a standard thrice-weekly HD schedule for longer than 5 years, and (iii) use of low-flux membranes. Exclusion criteria were (i) online HDF or HD with high-flux membranes, (ii) active malignancy or infection, or (iii) history of alcohol or drug abuse or known severe mental disorder. In the second part of the study, we included ESKD patients who were treated with standard HD, xHD, or online HDF. Inclusion criteria were (i) age >18 years and (ii) ESKD treated with a thrice-weekly HD schedule for longer than 5 years; and exclusion criteria were (i) active malignancy or infection and (ii) history of alcohol or drug abuse or mental disorder. All the participants were dialyzed with NIKKISO DBB EXA^®^ machines (Tokyo, Japan). As for the dialyzers, we used FX10^®^ in the standard HD group, FX100^®^ in the online-HDF group, and THERANOVA^®^ in the xHD group. The flow rate was between 300 and 400 mL/min, and the dialysate flow rate was 500 mL/min; calcium concentration was 1.75 mmol/L, magnesium concentration was 0.5 mmol/L, potassium concentration was 2 mmol/L, and sodium concentration was 139 mmol/L during the whole period of the study. Moreover, there was stable medical and nursing staff throughout the duration of the study. All the participants underwent three 4 h dialysis sessions per week. As for diet, all the participants were strongly advised to have a protein intake of 1–1.2 g/kg/day and a daily energy intake of 30–35 kcal/kg; they were also advised to strictly reduce salt consumption to <2 g/day. As for medication, no changes were allowed during the study period, and all the participants were strongly advised to receive their medication with no deviations. All the participants provided signed consent for their participation in the study after being thoroughly informed about the study procedures and potential risks and benefits. All protocol procedures were approved by the Ethics Committee of the University Hospital of Patras (approval code: 499/9-11-2021) and were conducted in accordance with the Declaration of Helsinki (2013 Amendment).

### 4.2. Study Procedures and Data Collection

Participants that fulfilled the inclusion/exclusion criteria and consented to participate after written informed consent were treated with xHD using appropriate standardized filters for 4 weeks. The effectiveness of xHD in the removal of low- (urea) and medium-molecular-weight (β2-microglobulin) substances was measured at the beginning and at the end of the study period and was compared with the standard HD treatment.

In the second part of our study, the aim was to compare the clearance of medium-molecular-weight substances like soluble receptor of urokinase plasminogen activator (Su-PAR) (molecular weight: 24–66 kDa), activin-A (molecular weight: 20 kDa), and monocyte chemoattractant protein-1 (MCP-1) (molecular weight: 11–13 kDa) between standard HD with low-flux membranes, online HDF, and xHD. For this purpose, we measured these molecules before and after HD in 10 patients on HD with low-flux membranes, 10 patients on online HDF, and 9 patients on xHD. Patients on the xHD arm were treated with medium cut-off (MCO) membranes for four weeks before the sample collection and were previously treated with standard HD with low-flux membranes.

### 4.3. Serum Su-PAR, Activin-A, MCP-1, and Standard Biochemical Measurements

All serum samples obtained from patients were appropriately stored at −80 °C until assayed after separation from clotted blood by centrifugation for 10 min at 1200× *g* in 4 °C. Serum Su-PAR, activin-A, and MCP-1 concentrations were measured using commercially available sandwich ELISA kits (Su-PAR: DUP00; activin-A: DAC00B; MCP-1: DCP00; R&D Systems, Minneapolis, MN, USA). All tests were performed according to the manufacturer’s recommended protocols. The participants’ serum urea, creatinine, and β2-microglobulin profiles were also determined using an automated analyzer (ADVIA^®^ 2400 Chemistry System, Siemens, Munich, Germany).

### 4.4. Statistical Analysis

All statistical analyses were performed with GraphPad Prism 8 software. Continuous variables are presented as mean with standard deviation (SD) or median and interquartile range (IQR) depending on the normality of distribution, which was assessed with the Shapiro–Wilk test and confirmed by visual assessment of curves in histograms. Categorical data are presented as frequencies and percentages (*n*, %). The comparisons between standard and xHD were performed using the paired *t*-test or the Wilcoxon matched-pairs signed-rank test in case of violation of normality. In order to compare the differences in Su-PAR, activin-A, and MCP-1 between the three groups, we used a one-way analysis of variance (ANOVA). A *p*-value level <0.05 was considered statistically significant.

## 5. Conclusions

In conclusion, this prospective non-randomized open-label study showed that xHD is very effective in the clearance of medium-molecular-weight uremic toxins, which could potentially confer a survival benefit if associated with the reduction in molecules such as MCP-1 and Su-PAR; however, a longer observation period is needed to clarify if this reduction in medium-molecular weight toxins could lead to a reduction in cardiovascular events and mortality. Further randomized controlled trials are warranted to fully elucidate the effects of xHD on the clearance of medium-molecular-weight substances and whether they could have an impact on morbidity and mortality associated with HD.

## Figures and Tables

**Figure 1 ijms-26-05747-f001:**
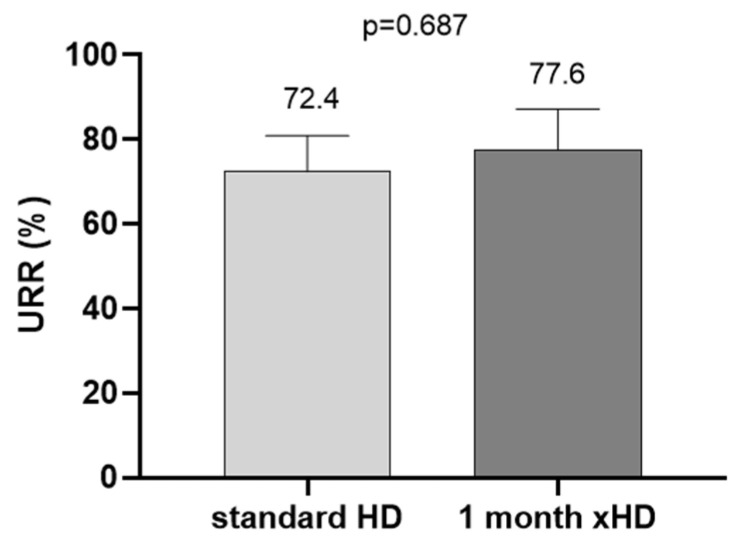
URR (%) at standard HD and after 4 weeks on xHD.

**Figure 2 ijms-26-05747-f002:**
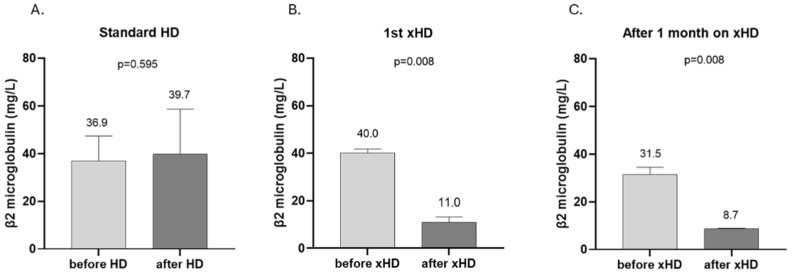
β2-microglobulin concentrations before and after HD (**A**), the first xHD session (**B**), and one month after xHD implementation (**C**). (**A**), mean ± SD; (**B**,**C**), median, IQR.

**Figure 3 ijms-26-05747-f003:**
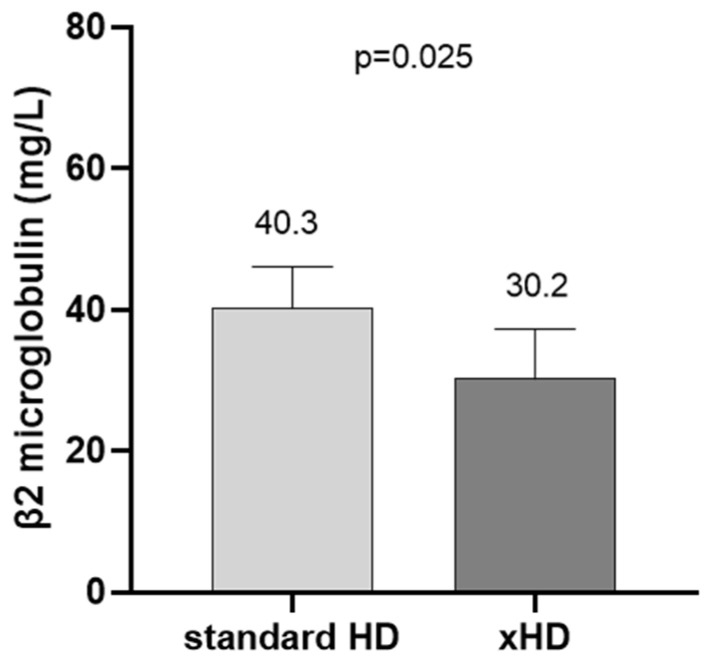
β2-microglobulin levels before HD session at baseline and after 4 weeks on xHD.

**Figure 4 ijms-26-05747-f004:**
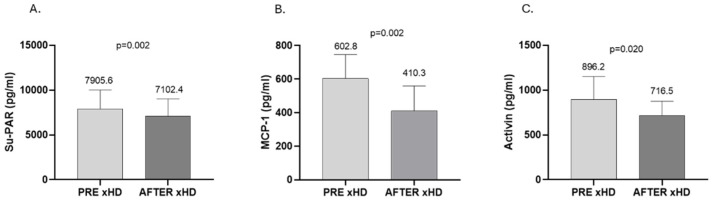
Effect of xHD on the clearance of Su-PAR (**A**), MCP-1 (**B**), and activin (**C**). (**A**,**C**), mean ± SD; (**B**), median, IQR.

**Figure 5 ijms-26-05747-f005:**
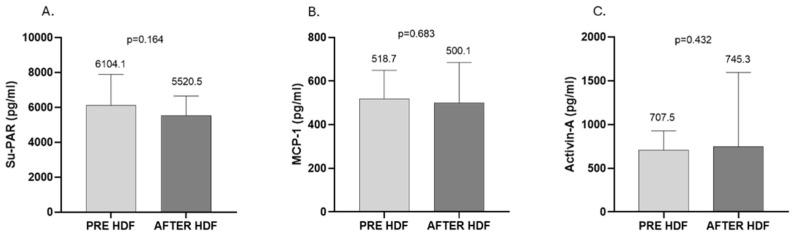
Effect of online HDF on the clearance of Su-PAR (**A**), MCP-1 (**B**), and activin (**C**). (**A**,**B**), mean ± SD; (**C**), median, IQR.

**Figure 6 ijms-26-05747-f006:**
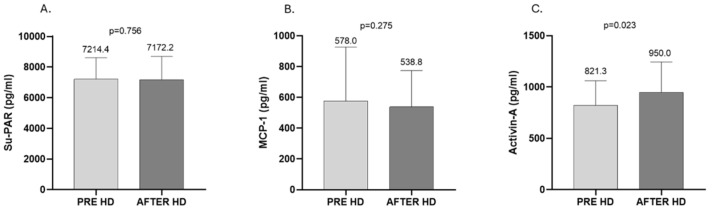
Effect of standard HD on the clearance of Su-PAR (**A**), MCP-1 (**B**), and activin (**C**). (**A**,**C**), mean ± SD; (**B**), median, IQR.

**Table 1 ijms-26-05747-t001:** Patients baseline demographic and clinical characteristics.

(**A**)			
*n*			7
Male (*n*, %)			4, 57%
Age (y)			64.2 ± 10.8
Hemodialysis vintage (y)			10.8 ± 4.6
Baseline Renal Disease			
Diabetic Nephropathy (*n*, %)			2, 28.6%
Hypertensive Nephrosclerosis (*n*, %)			3, 43%
Glomerulonephritis (*n*, %)			2, 28.6%
Vascular access			
AVF (arteriovenus fistula) (*n*, %)			4, 57%
AVG (arteriovenus graft) (*n*, %)			1, 14.3%
Central catheter (*n*, %)			2, 28.6%
Comorbidities			
Diabetes mellitus (*n*, %)			2, 28.6%
Hypertension (*n*, %)			7, 100%
Coronary artery disease (*n*, %)			2, 28.6%
Heart failure (*n*, %)			1, 14.3%
(**B**)			
	**Standard HD**	**Online HDF**	**xHD**
*n*	10	10	10
Male (*n*, %)	7, 70%	7, 70%	7, 70%
Age (y)	61.7 ± 5.3	61.7 ± 4.9	61.9 ± 6.0
Hemodialysis vintage (y)	6.1 ± 1.4	5.8 ± 1.3	5.8 ± 1.3
Baseline Renal Disease			
Diabetic nephropathy (*n*, %)	3, 30%	4, 40%	3, 30%
Hypertensive Nephrosclerosis (*n*, %)	4, 40%	3, 30%	30, 30%
Glomerulonephritis (*n*, %)	3, 30%	3, 30%	4, 40%
Vascular access			
AVF (arteriovenus fistula) (*n*, %)	6, 60%	7, 70%	6, 60%
AVG (arteriovenus graft) (*n*, %)	2, 20%	2, 20%	2, 20%
Central catheter (*n*, %)	2, 20%	1, 10%	2, 20%
Comorbidities			
Diabetes mellitus (*n*, %)	4, 40%	4, 40%	4, 40%
Hypertension (*n*, %)	10, 100%	9, 90%	10, 100%
Coronary artery disease (*n*, %)	2, 20%	2, 20%	1, 10%
Heart failure (*n*, %)	1, 10%	0, 0%	1, 10%

## Data Availability

All anonymized data related to this work are available upon request to the corresponding author.

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
