# Peer review of "Efficacy of Expanded Hemodialysis and Comparison with Standard Hemodialysis and Online Hemodiafiltration"

_ijms, 2025, doi:10.3390/ijms26125747_

Round 1

Reviewer 1 Report

Comments and Suggestions for Authors
  1. The first part of the study included only 7 patients, and the small sample size may have led to insufficient statistical power, affecting the stability and generalizability of the results. It might not have adequately captured individual differences, so the conclusions may be uncertain when applied to a wider population.
  2. With only a 4 - week intervention period, the study may not have been long enough to fully assess the long - term efficacy and safety of the dialysis methods. Long - term complications like chronic inflammation and cardiovascular disease require more prolonged observation to evaluate their development and the impact of dialysis.
  3. The study used a non - randomised open - label design, which can introduce selection bias and confounding factors. A randomised controlled trial design is recommended for more objective results in future studies.
  4. Data from a single centre may be influenced by factors specific to that environment, such as equipment and staff, which could affect the results and limit their generalisability.
  5. Beyond basic patient characteristics, other factors that might affect dialysis outcomes, such as diet, medication use, and dialysis adherence, were not recorded or analysed in detail. These factors could interfere with the assessment of dialysis methods' effectiveness.
  6. The study showed that xHD reduces certain middle - molecular - weight toxin levels, but it's unclear if these biochemical changes lead to actual clinical improvements like fewer cardiovascular events or lower mortality.
  7. The study states that ethical approval and patient consent were obtained, but it doesn't go into detail about the consent process, patients' rights, and privacy protection.

Reviewer 2 Report

Comments and Suggestions for Authors

This study provides promising evidence for xHD as an efficient, cost-effective method for middle molecule clearance. However, methodological limitations and small sample sizes necessitate cautious interpretation. Addressing these issues in revisions would strengthen the paper’s validity and impact.

Major Concerns

  1. Sample Size and Study Duration:

   - Part 1 included only 7 patients, and Part 2 had 10 patients per group. Small sample sizes reduce statistical power and generalizability.

   - The 4-week intervention period is insufficient to assess long-term efficacy or clinical outcomes (e.g., mortality, cardiovascular events).

  1. Non-Randomized Design:

   - Lack of randomization introduces selection bias. Patients were not randomly assigned to xHD/HDF/sHD groups, potentially confounding results.

  1. Statistical Analysis:

   - Paired t-tests/ANOVA were used without addressing non-normality or multiple comparisons (e.g., Bonferroni correction). Non-parametric tests (e.g., Wilcoxon) are more appropriate for small samples.

  1. Ethical and Reporting Gaps:

   - Missing Institutional Review Board (IRB) approval statement and incomplete informed consent details.

  1. Overinterpretation of Results:

   - The conclusion speculates on "survival benefit" without measuring survival or long-term outcomes. Biomarker reductions are not directly linked to clinical endpoints.

Reviewer 3 Report

Comments and Suggestions for Authors

The present study addresses the effectiveness of different forms of hemodialysis on the clearance of middle molecules, a question that has already been examined in much more comprehensive studies. The small cohort considered here does not yield any unexpected results for the reader. Currently, the manuscript exhibits significant limitations, starting with the poorly described methodology, which lacks clear information about the materials used. Without this information, the few presented results cannot be adequately interpreted. The major limitations listed below must be addressed:

Title and Abstract: The title of the manuscript does not correspond to the abstract, as the term 'hemodiafiltration' is not mentioned at all in the abstract. This discrepancy needs to be addressed to ensure consistency.

Additionally, the use of the abbreviation 'expanded HD' (xHD) is inconsistent throughout the abstract. A unified terminology should be maintained.

Introduction: The statement 'has made great advances' is overly exaggerated. Dialysis remains an inferior solution compared to transplantation. The statement should be moderated and rephrased accordingly.

The claim of a direct and singular causality between middle molecular weight toxins and conditions such as amyloidosis, chronic inflammation, cardiovascular disease, mineral-bone disease, and neuropathy is not scientifically sound. These toxins may contribute to these conditions, but presenting a monocasual relationship is inappropriate.

Methods: The methods section lacks essential details, specifically concerning the dialysis machines and filters used. Information about filters is critical as variations could explain the study’s findings, given the known differences between various filter types.

There is a lack of clarity regarding the term 'expanded HD,' including the duration of the hemodialysis sessions and the specific filters used in each group.

Regarding study design, a group of patients transitioning from standard HD to online HDF would have provided more insightful results. The relevance of standard HD with low-flux filters is questionable, as it is considered outdated.

Results: The results section lacks a baseline table including essential statistics (age, dialysis vintage, baseline renal disease, dialysis access). Providing such a table is a scientific standard, especially considering the small patient cohort.

The labeling of groups is confusing: In Figure 1, 'baseline' is labeled as 'standard HD,' while in Figures 2B and 2C, 'before HD' and 'after HD' would be more accurately labeled as 'before xHD' and 'after xHD.'

The statement regarding the comparison of Su-PAR, MCP-1, and activin-A clearance between the groups lacks a corresponding figure. Figure 4 only shows pre/post xHD comparisons. Please provide data for all groups, including percentage changes.

Discussion: The statement that expanded HD could be a reliable and cost-effective alternative to on-line HDF requires clarification. Information on cost differences and practical feasibility, particularly in routine patient care, is necessary. In most dialysis centers, optimizing patient numbers is a priority, making expanded HD potentially less practical.

The 2023 landmark study (CONVINCE study by Peter Blankestijn, DOI: 10.1056/NEJMoa2304820) demonstrated improved mortality with HDF compared to high-flux HD. The authors should address this study and incorporate its findings into the discussion.

Conclusion: The final statement claiming that further randomized trials are necessary to evaluate the effects of MCO membranes on medium molecular weight substance clearance is misleading. The study lacks focus on MCO membranes, and more specific information regarding the methodology is required.

General Comments: The study’s relevance is questionable, given the established knowledge among nephrologists that HD with low-flux filters is outdated. This fundamental issue should be addressed.

Round 2

Reviewer 2 Report

Comments and Suggestions for Authors

No further comments, it can be published.